# Guardians of the Text: Griffins and Sphinxes in the Neapolitan Ovid (BNN ms. IV F 3)

**Fátima Díez-Platas**

Department of History of Art, Universidade de Santiago de Compostela, 15782 Santiago de Compostela, Spain;
fatima.diez@usc.es

**Abstract:** This article investigates the origins and significance of images of griffins and sphinxes—hybrid creatures of Greco-Roman tradition—in the marginal decorations of the so-called "Neapolitan Ovid" (BNN ms. IV F 3), the first illuminated manuscript of the *Metamorphoses*, probably from the late 11th century. Their form and style suggest specific iconographic origins and links with the decorative motifs from Antiquity that circulated in artistic objects around Bari, the manuscript's place of origin. Among the many figures that provide a pictorial response to the poem's content, the presence of these griffins and sphinxes offers compelling evidence of the survival of ancient imagery; they also invite us to explore the relationship between image and text in the illuminated book. From this analysis, we can better understand the complex role of these hybrid figures in the manuscript. Their existence is testimony to the continuity of marginal decorative systems derived from Antiquity that are present in objects and were articulated through the Islamic and Byzantine formal vocabularies accessible in Puglia at the time.

**Keywords:** *Metamorphoses*; miniature; Islamic arts; Byzantine arts; Bari; Puglia; Orientalizing motifs; hybrid figures; lion; panther

## 1. Introduction

The manuscript of Ovid's *Metamorphoses*, known as the "Ovidio Napoletano" (Biblioteca Nazionale Vittorio Emmanuele III, Ms. IV F 3),[1] copied and decorated in Bari probably at the end of the 11th century,[2] is an exceptional codex that is regarded not only as one of the key versions for the reconstruction of the original Ovidian text (Magistrale 1998, pp. 46–51), but also as the oldest surviving illuminated manuscript of the Ovidian work.[3] With its unique decoration, the iconic apparatus of this early illustrated version of the *Metamorphoses* constitutes a new and extraordinary visual response to Ovid's text, the poem of change, whose rapid shifts of meaning and suggestions of form and color define the nature created by the poet and open the way to the fiction of art.

The lavish decoration of the "Napoletano" features an outstanding collection of different images, both in number and variety, arranged in the margins of the page. As a whole, the figurative apparatus constitutes a special type of illustrative device consisting of various kinds of vivid and colorful human, hybrid, and animal figures alternating with other visual forms. These elements include thirteen decorated capitals from books II to XIV,[4] pseudo-Cufic letters and *rumi* leaves that accompany and complement the numerous forms and different shapes of the summaries of the *narrationes* attributed to Pseudo-Lactantius, which were enclosed in decorated miniature frames after the completion of the copy (Magistrale 1998, p. 71). Of the two hundred and one folios that currently make up the manuscript, one hundred and sixty-eight pages are decorated with figurative miniatures, vegetal ornaments, and the aforementioned "visual summaries", for a total of one hundred and three miniatures made up of one hundred and twenty different figures, plus the one hundred and fifty-seven textual images that make up the Lactantian *narrationes*. This is a complex visual system that raises questions about its meaning and function, since

the *mise-en-page* of the manuscript reveals a purposeful relationship between the various decorative devices and the text, which is also decorated. The special layout is composed of the figurative ensemble of animated figures, the phytomorphic and zoomorphic elements that develop from the geometric forms of the frames of the summaries, and the embellished text of the poem, which is characterized by a rich initial ornamentation: intricate figurative book capitals, the enlarged and colored initials that indicate the beginning of each *fabula*, and the colored initials of each verse (Figure 1).

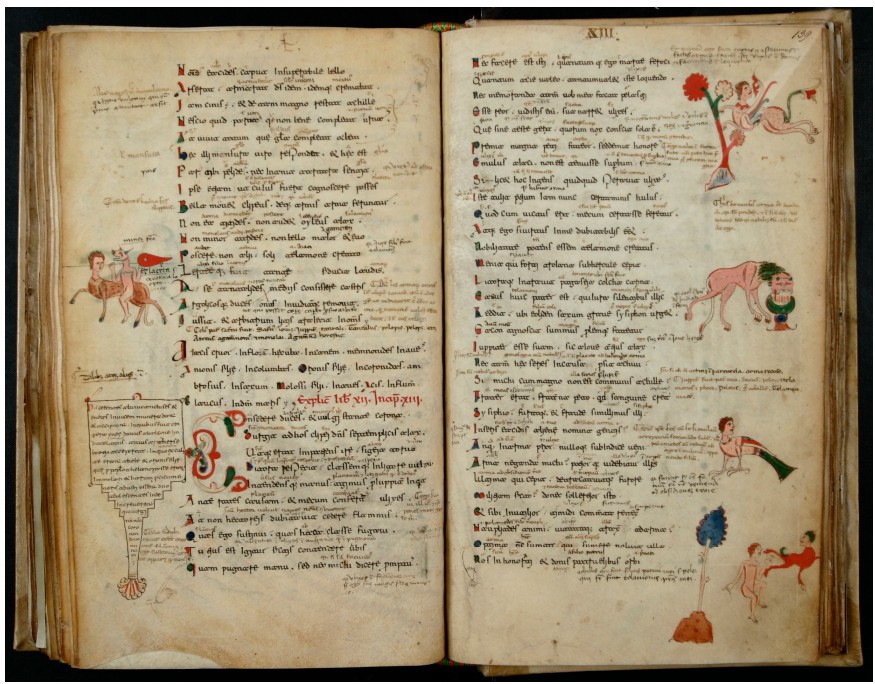

**Figure 1.** The Neapolitan Ovid (BNN ms. IV F 3), fols. 158v–159r. (Public Domain).

The attention paid to the artistic and visual aspects of the Bari manuscript has been mainly carried out by Carlo Bertelli, who was the first scholar to point out the exceptional aspects of the codex decorations (Bertelli 1975, 1978, 1983, 1987), and by Giulia Orofino, who has devoted herself intensively to analyzing and interpreting the complex figurative system from both an iconographic point of view and in terms of stylistic influences (Orofino 1993, 1995, 1998, 2020).[5] Beginning in 1975, Bertelli drew attention to the illumination of the manuscript and provided a summary description of the decoration, revealing the connection between the lively figures and the ornamentation of the pages with the decorative repertoire of motifs found on objects of Byzantine and Islamic origin and in the art of 11th- and 12th-century Puglia (Bertelli 1975, p. 925). On the basis of these intuitions, Orofino has addressed a brilliant revision of the formal aspects of the manuscript and the miniatures as well as of the models, parallels, and influences of the heterogenous figurative ensemble. Regarding the relationship between image and text in the manuscript, she has considered this first attempt of illustration of the *Metamorphoses* as a particular way of translating the content of the poem into images, attributed to the initiative of a single artist of extraordinary imagination and exuberant creativity. This approach offers a unique interpretive version of the mythological text that produces "in margine a quello ovidiano il romanzo medievale della mitologia"[6] (Orofino 1993, pp. 6–7).

In this sense, the purpose of this contribution is to examine particular aspects of the visual apparatus of this manuscript with the intention of carrying out an iconographic and functional analysis that seeks to complement the one previously carried out by Orofino and the recent detailed overview by Vandi (2019). On this occasion, the approach focuses on the conspicuous appearance of griffins and sphinxes in the margins of the pages of

the "Napoletano", a feature that offers an extraordinary opportunity to further explore the complex structure of its figurative system.

## 2. Sorted Images: The Decorative System of the Neapolitan Ovid

> "Esas ambigüedades, redundancias y deficiencias recuerdan las que el doctor Franz Kuhn atribuye a cierta enciclopedia china que se titula *Emporio celestial de conocimientos benévolos.* En sus remotas páginas está escrito que los animales se dividen en (a) pertenecientes al Emperador (b) embalsamados (c) amaestrados (d) lechones (e) sirenas (f) fabulosos (g) perros sueltos (h) incluidos en esta clasificación (i) que se agitan como locos (j) innumerables (k) dibujados con un pincel finísimo de pelo de camello (l) etcétera (m) que acaban de romper el jarrón (n) que de lejos parecen moscas". Jorge Luis Borges, "El idioma analítico de John Wilkins", *Otras Inquisiciones*, 1952[7]

The overall figurative system of the miniatures of the Neapolitan Ovid includes human figures, animals, and hybrids, a series of images of different types that seem to have different meanings. First, there are human figures in various poses and garments, some of them nude; second, there are hybrid figures of various compositions; and third, there are numerous animals of different types and species in identifiable poses, such as dogs, birds, cows, bears, hares, a camel, a strange boar, and several felines, some of which form incipient scenes that can be read as typical or evocative of hunting or play.

The text that introduces this section, a fragment of the classification of animals from a fictional encyclopedia alluded to by Jorge Luis Borges in a story about the organizing capacity of language, seems an adequate description of the apparently chaotic set of images found in the Ovidian manuscript. In it, it is precisely the fabulous hybrids, such as griffins and sphinxes, a pair of sirens or harpies, and the numerous unleashed dogs, that attract our attention. They are all part of the group of figures that pose a problem in the task of identifying the miniatures of the Neapolitan manuscript as illustrations of the Ovidian poem.

Thus, the interpretation of the decoration of the manuscript as the first illustration of the *Metamorphoses* has turned out to be an arduous and almost disappointing task, above all because it is not easy to recognize the characters of the Ovidian stories in the motley array of figures that swarm over the pages of the manuscript. On the one hand, Giulia Orofino's attempt to establish the relationship of the various figures and incipient scenes to the text, or rather to the content of the poem, has led her to the conclusion that the figurative proposals related to the content do not respond to classical visual models, but are rather constructed as purely "textual" products using the text as the only source for elaborating their visual incarnation. On the other hand, she has suggested that the vast majority of the figures that illustrate the text are not actually literal images, but are instead products of the artist's own creative version stimulated by the Ovidian poetry itself (Orofino 1993, p. 6).

First of all, there are certain figures in which the reflection of mythological stories is clearly recognizable: the figure of Argos (fol. 16r; Figure 2), whose body is covered with eyes, which looks like a figure from the ancient Greek pottery (Figure 3); the abbreviated zodiac running from the top right margin to the bottom of the page (fol. 20r), which refers to the vision of Phaethon falling from the sky when Zeus knocks him down from the chariot of the sun; the winged figure of a naked boy sitting on a structure, which seems to represent Icarus about to take flight (fol. 101r); the hunting scene of a stag running through the pages (fols. 14v–15r; Figure 4), which refers to the tragic end of the metamorphosed hunter Actaeon; and the lion drinking from a fountain, which seems to be an incipient scene detached from the complex story of Pyramus and Thisbe (fol. 159r; Figure 1).

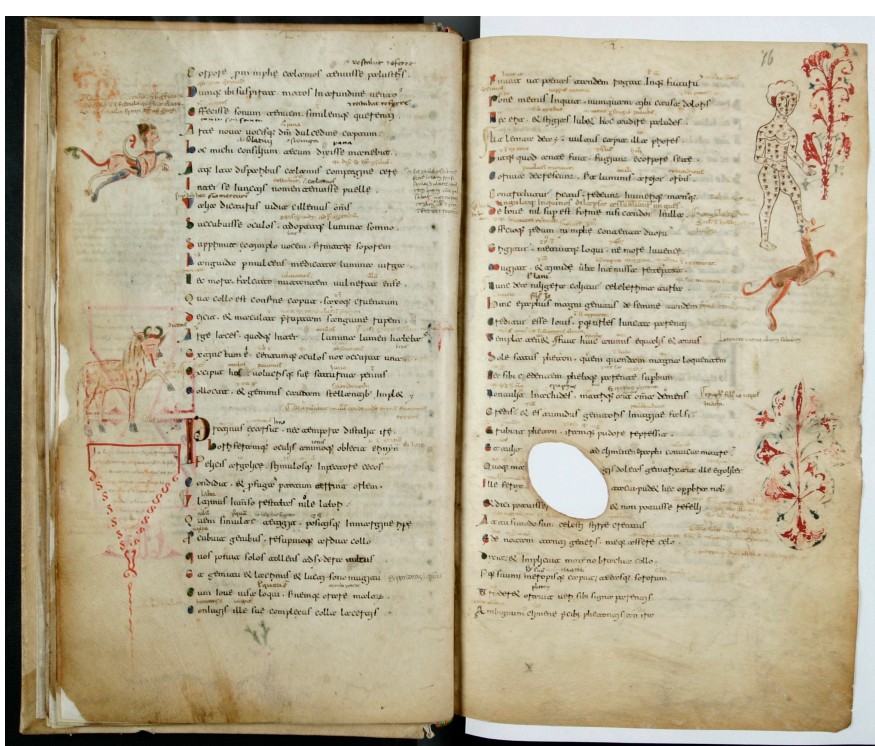

**Figure 2.** Neapolitan Ovid (BNN ms. IV F 3), fols. 15v–16r. (Public Domain).

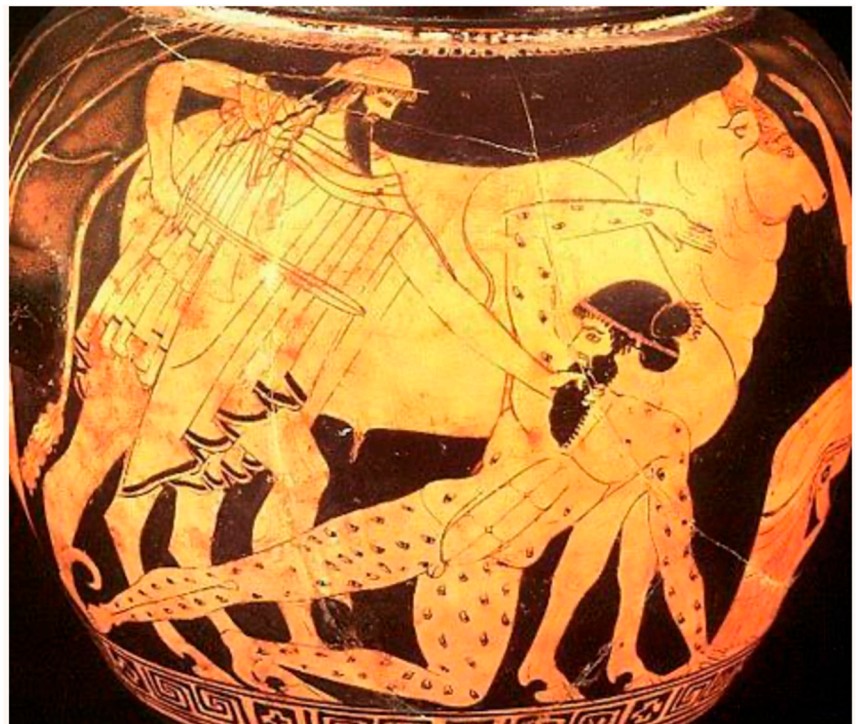

**Figure 3.** Hermes killing Argos *panoptes*. Red figure stamnos. From Cerveteri. Ca. 460 BC. Vienna, Kunsthistorisches Museum IV 3729. Photo: *Theoi Greek Mythology:* https://www.theoi.com/Gallery/L11.2.html, accessed on 5 March 2023.

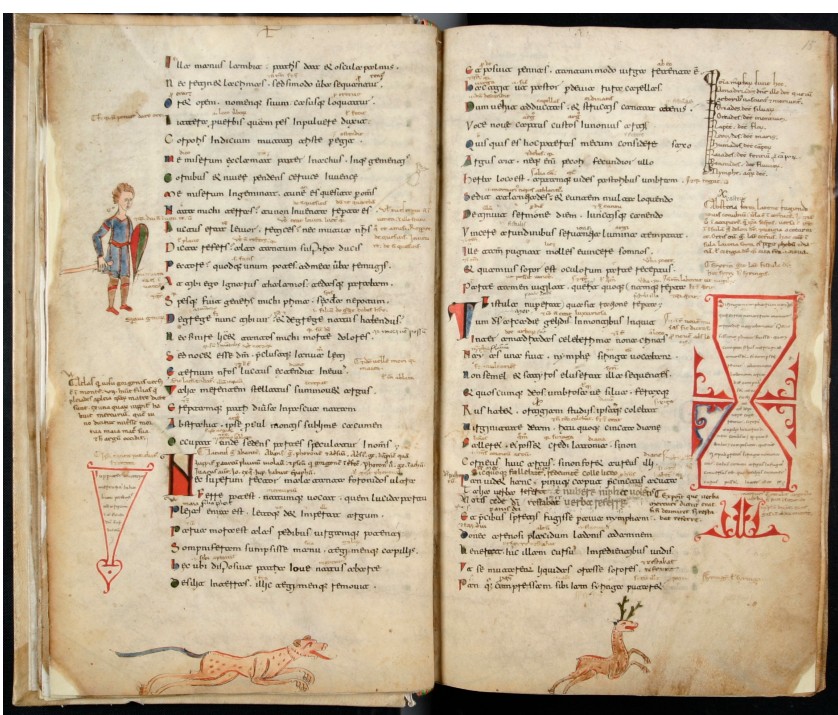

**Figure 4.** Neapolitan Ovid (BNN ms. IV F 3), fols. 14v–15r. (Public Domain).

Second, among the animal images that populate the manuscript, the figures of two cows (fols. 15v: Figure 2; and fol. 102v) and of two bears (fol. 36v: Figure 5; and fol. 71v: Figure 6) respectively evoke the stories of Io and Callisto through the results of their metamorphoses, just as the images of isolated trees next to the text (fol. 16r: Figure 2; fols. 21v and 22r) could refer to the transformation of Daphne into a laurel or of the Heliades, Phaethon's sisters, into weeping willows (Orofino 1993, p. 8). This individualized way of representing animals or plant elements seems to be close to the way in which contemporary bestiaries and encyclopedias are illustrated (Díez Platas 2021, pp. 460–63).

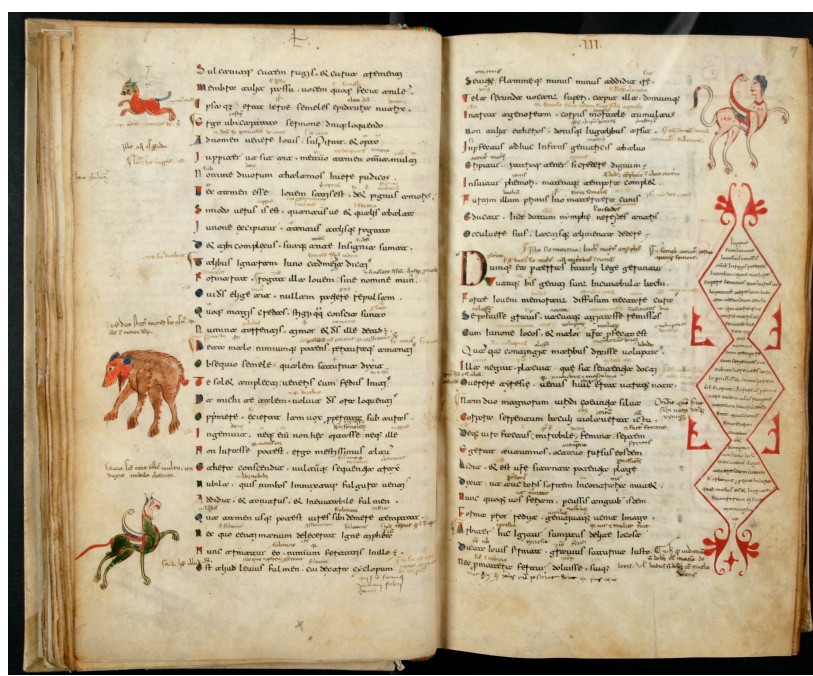

**Figure 5.** Neapolitan Ovid (BNN ms. IV F 3), fols. 36v–37r. (Public Domain).

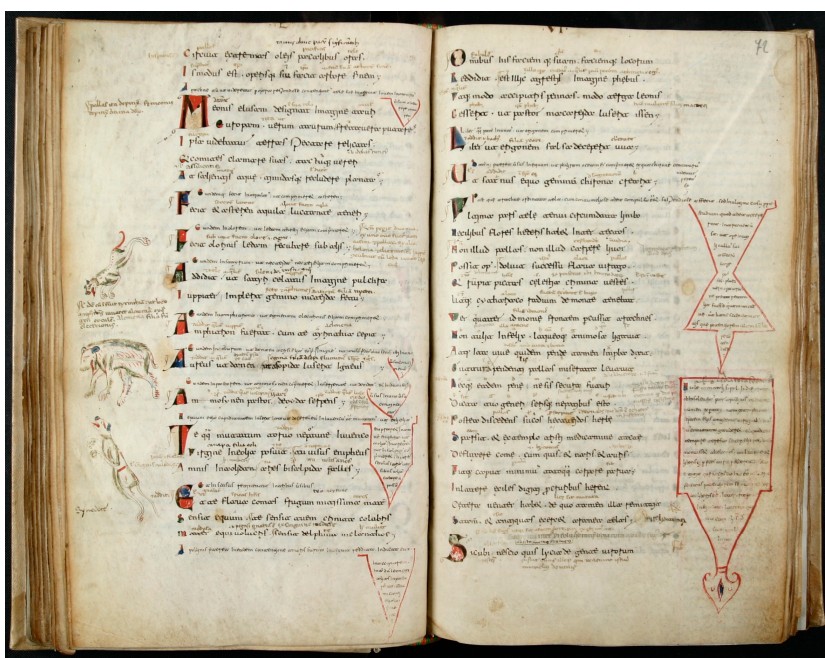

**Figure 6.** Neapolitan Ovid (BNN ms. IV F 3), fols. 71v–72r. (Public Domain).

It is possible to discern a visual version of some of the poem's stories in medievalized proposals that depart from classical formulas depicted next to the stories that describe them (Orofino 1993, p. 6). Thus, a kind of sea dragon derived from a known model and identifiable with the image of a *Cetus* accompanies the text of the transformation of Cadmus into a snake (Book IV, fol. 54r). The minotaur (Book VIII, fol. 101r) is presented in his medieval guise as a kind of taurine centaur, with mottled skin and a human head that is also female (Díez Platas 2005), and the sorceress Circe (Book XIIII, fol. 176r) is a naked female figure riding a goat with strange horns (Orofino 1993, p. 6; Castiñeiras 2020). Finally, as Orofino has pointed out, in certain figures in the manuscript we might be able to recognize the process of metamorphosis itself, as in the figure of the contortionist with a flexible, elongated, and colored waist depicted at the point of throwing himself backward in an acrobatic movement. This image, which seems to reproduce literally the words of Ovid ("Terga caput tangent, colla intercepta videntur, spina viret, venter, pars maxima corporis, albet" *Met.* VI, 379–81), represents, according to Orofino (1993, p. 8), one of the Lycian peasants who refuses to help Latona and is consequently transformed into a frog.

However, the hope of recognizing the stories of the poem in the figures depicted in the manuscript is hampered by a second obstacle, in that it is even more difficult to associate most of the recognizable figures mentioned above with the verses next to which they are depicted. Except for a few figures, most of the recognizable images are not shown next to the verses that describe them. Furthermore, neither the bear, which recalls Callisto, nor the stag, which recalls the tragic end of Actaeon, appear on the pages where the texts of these stories are found.

This fact, which represents a real "dislocation" of the illustration, has had, in my opinion, two different effects. On the one hand, it has stimulated the search for the meaning of certain figures in relation to the text next to which they are represented. Thus, Orofino resorts to metaphor and evocation in order to read the figures of the camel next to the story of Narcissus (Book III, fol. 39v) and the group of eagles that fall on a hare and prey on it next to the description of the pursuit of Scylla by her father Nyssus (Book VIII, fol. 100r), both transformed into birds, as hints of the artist's imaginative interpretation (Orofino 1993, pp. 8–9).

On the other hand, the displacement of the figures provides the impression that the figures in the manuscript are free, as if they had been released from the interior of the text,

as already noted by Bertelli (1975, p. 925) and corroborated by Orofino (1993). Thus, from my point of view, the figures are not "tied" to the text, nor are they meant to be mere concrete illustrations that must be read with the text in order to be understood. I would argue that the figures, recognizable as the transformed forms produced by the poem, function as mnemonic devices or cross-references that are identified as one reads along in the flow of the fable. In this sense, Orofino's intuition becomes real in relation to the liveliness and freedom of the visual response produced by the miniaturist, who allows himself to be infected by the spirit and fluidity of the Ovidian story itself (Orofino 1993, p. 8). The figures seem to run through the manuscript, moving back and forth as if inside a game board that the manuscript becomes, a sort of image container (to which we will return later).

Thus, having exhausted the identification of the recognizable images that function as visual translations of the narrative stories and having explored the possible evocations that can be linked to a metaphorical reading of Ovidian poetry, it remains to explain the role of the large number of images without identifiable meaning, such as the apparently decorative animals and hybrids that circulate freely in the margins of the manuscript and cannot be linked to the stories or allusions of the poem. The interpretation of some of the animals in the light of the bestiaries, such as the camel glossing the story of Narcissus, emphasize the attitudes and moral qualities of the animals and show their obvious influence; however, this does not explain the presence of numerous dogs, lions, and leopards or panthers that are clearly unrelated to the verses that run close to them. Second, we must face the problem of the numerous hybrid creatures, which only in a few cases seem to be responsible for any representation of the metamorphoses,[8] leaving the numerous griffins and sphinxes as meaningless images in relation to the illustration of the text.

Therefore, with rare exceptions, the images in the margins of the manuscript do not seem to form a coherent and consistent structure with the content of the text; that is, they do not really function as illustrations, and they lack a clear syntax that can be explained by the content of the poem. Furthermore, the extreme individualization of the images reinforces the impression that the relationship between them is entirely paratactic, making them even more unrelated, deeply individuated and almost isolated figures, as if they were not part of a program or a coherent system.

Moving on to another level of interpretation, the variety of forms and figures presents the impression that each has no clear purpose in relation to the text or its content. Lacking or being deprived of an obvious relation to the stories, the figures seem condemned to serve only a decorative purpose, deprived of the task of conveying any meaning other than the one revealed by the style, the form, and obvious liveliness of the execution. In the end, they seem to be reduced to a constellation of images that serve only to embellish the page and the manuscript.

However, a more detailed analysis, leaving aside the problem of the literal translation of the content and the question of the spatial association of the verses and figures within the manuscript, can allow us to identify a kind of underlying syntax that manifests itself in the fluidity of the image and in the awareness of its relationship to the text rather than to the content of the poem. This syntax goes beyond the page and the *mise-en-page*, building a structure that runs through the whole manuscript, which must be conceived of as a book, and requires that the sequence of pages and sequence of time be openly manifested through the pictorial space, that is, in the images that literally run through the pages. One example of this can be found in the Actaeon scene (fols. 14v–15r; Figure 4), as previously noted by Vandi (2019, p. 314).

In this way, it seems to me that it is possible to establish various relationships between some of the images and the manuscript, and between some of the images and the text, in order to identify a certain structure designed to show the relationship with the content of the poem as well as with the references to Antiquity and the repertoire of figures or models through which the illuminator has composed this alternative Ovidian world that conveys a mediated image of the classical past. Therefore, I would like to pay special attention to those figures that seem to have less meaning or to be relegated to a marginal position in the

decoration, and especially to the phenomenon of repetition, as there are groups of forms that seem to be united by their types of recurrent representation.

### 3. From the Ancient World: Griffins and Sphinxes in the "Napoletano"

In her brief review of the illumination of the Ovidian codex, Carla Lord has already drawn attention to the presence of hybrids "extraneous to the *Metamorphoses*" among the "images on the edge" (Lord 2011, p. 259). Although she mentions "centaurs, griffins, and harpies," the fact is that the most striking feature in the decoration of the manuscript is the recurring presence of images of griffins and sphinxes on the page margins. Indeed, it is the recurrence and formal unity of the figures of both these fabulous creatures as depicted in the Neapolitan Ovid that gives the impression that they could form a defined systematic group, suggesting a separate decorative level that conveys iconographic information.

There are eleven figures of griffins (fols. 3v, 16v, 29, 36v, 67, 76, 102, 116, 157v, 169v, 176v) and ten figures of sphinxes (fols. 11, 15v, 29, 37, 81v, 102, 104, 109, 125, 159, 180), which are consistent figures that could be formally defined as "canonical" because they keep the form of these mythological beings in the Greco-Roman tradition. All twenty-one figures in this individualized collection are unified by certain formal decorative features, such as vegetalized forms—especially noticeable in the shapes of the wings and tail—that are shared with a few other figures in the decorative set of images in the manuscript. However, despite the apparent stylistic unity of all the miniatures, as pointed out by Orofino (Orofino 1993, p. 5) there are certain differences between the consistent figures of what can be defined as the group of griffins and sphinxes[9] and the rest of the hybrid beings, and these seem to represent exercises in metamorphosis in response to the inspiration of the poem (See note 8 above) (Figure 1).

Nonetheless, despite the stylistic and formal rules that unite the griffins on the one hand and the sphinxes on the other (as well as both groups at the same time), all of the figures in both collections are truly individual, and each is different from the other (Figures 7–9). This outstanding feature confirms what has already been said about the creative and adaptive artistic capacity of the illuminator, who did not limit himself to reproducing the established and standardized models of each of the hybrids, and rather, while maintaining the recognizability of the typical figure of the hybrid creature, explored different gestures, postures, and colors for each of the instances (although always within a limited catalogue, as postures along with certain gestures and colors are used constantly and repeatedly in all the illuminated figures throughout the manuscript).

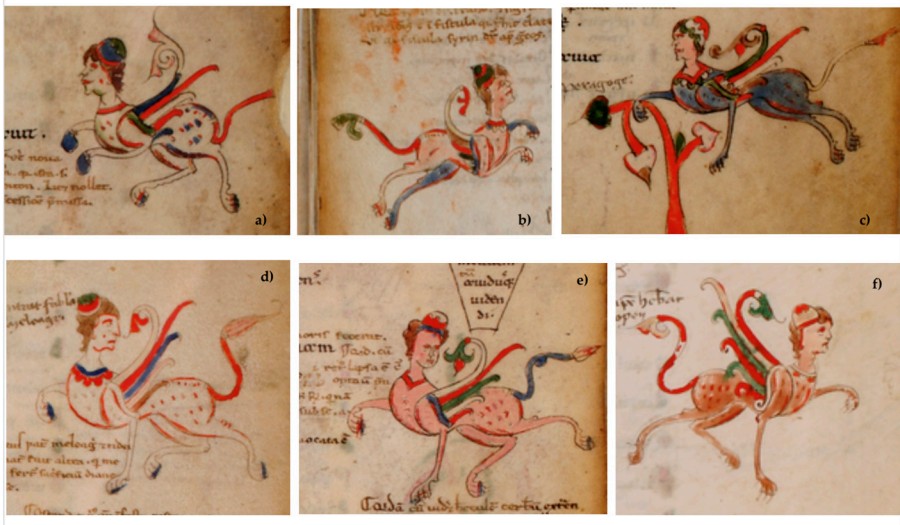

**Figure 7.** Neapolitan Ovid. Head-dressed sphinxes: (**a**) fol. 11r; (**b**) fol. 15v; (**c**) fol. 29r; (**d**) fol. 102r; (**e**) fol. 125r; (**f**) fol. 180r.

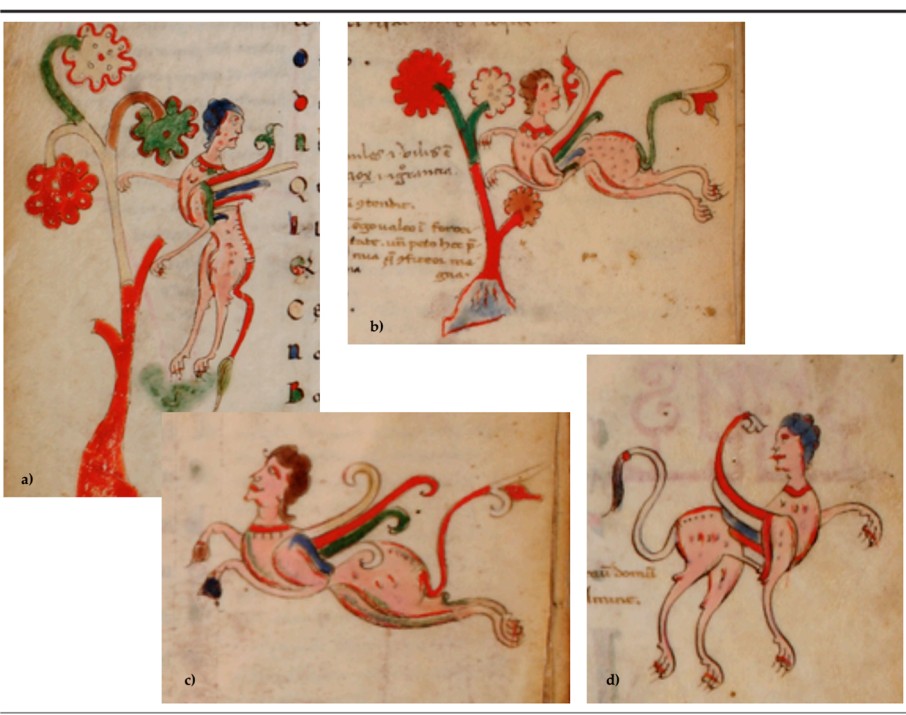

**Figure 8.** Neapolitan Ovid. Sphinxes with no headdress: (**a**) fol. 81v; (**b**) fol. 159r; (**c**) fol. 109r; (**d**) fol. 37r.

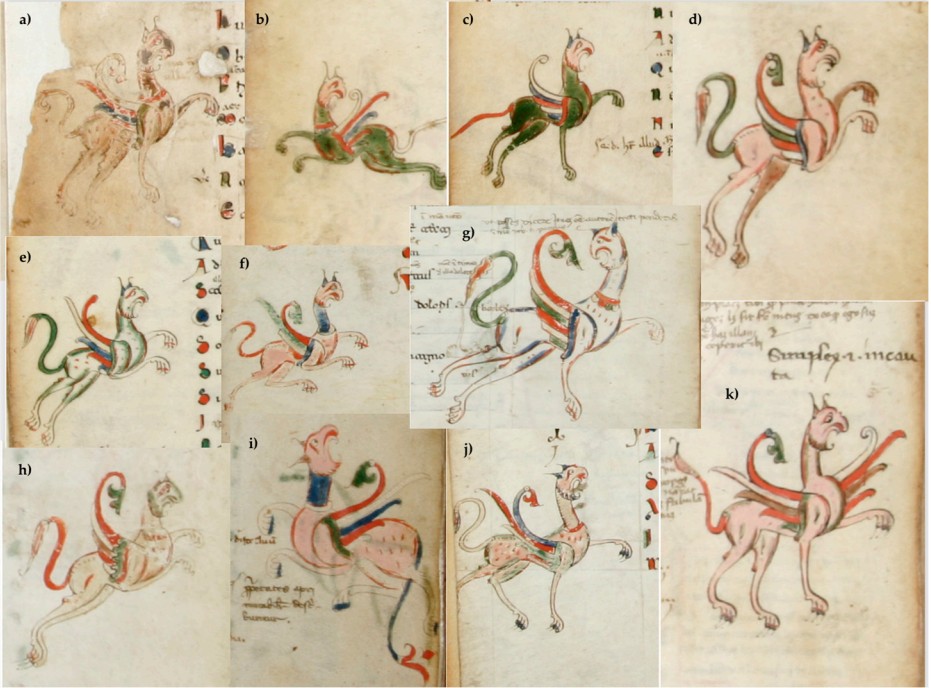

**Figure 9.** Neapolitan Ovid Griffins: (**a**) fol. 3v; (**b**) fol. 16v; (**c**) fol. 36v; (**d**) fol. 29r; (**e**) fol. 176v; (**f**) fol. 169v; (**g**) fol. 116r; (**h**) fol. 76r (**i**) fol. 102r; (**j**) fol. 157v (**k**) fol. 67r.

On the one hand, with regard to the sphinxes, which in their Greek configuration of clearly Oriental origin are always and consistently female, they were clearly conceived as a mixture of a woman and a feline, usually a lion, which in its most original and ancient version has wings.[10] However, this mixture is not always balanced, and varies from sphinxes with a female head and a winged lion's body to those with a female torso embedded in

a lion's body, and may or may not have wings. All ten sphinxes in the Neapolitan Ovid have a human head embedded in a winged lion's body, and only in a few cases does the composition seem to be altered by introducing parts of a bovid into the mixture, as can be seen in the hooves of the sphinxes in fols. 11r and 109r (Figures 7a and 8c). In addition, half of the sphinxes wear headdresses (Figure 7), while the other half do not and feature only a female head with short hair (Figure 8).

The griffins, on the other hand, are usually depicted in a much more stable form, corresponding as they must to the fusion of the most prominent animals of creation: the eagle, which provides the head and wings, and the lion, which provides the body base of the hybrid.[11] Although in certain medieval versions of the griffin the proportion of the eagle is increased such that the forelegs of the hybrid become the claws of the bird of prey, here all of the griffins are complete lions or winged felines with an eagle's head (Figure 9).

In characterizing these griffins and sphinxes, I would like to emphasize the particular interest in marking the value and importance of the head as a specific aspect in the conformation of their figures. In the Mycenaean world, sphinxes are recognizable from ancient times by their headdress and the way their hair is styled (Luján et al. 2017, pp. 450–55). While the value of certain features has not been studied in depth, it is likely that in addition to marking the status of intermediary beings between the divine and human worlds, they have the aim of highlighting the feminine condition of the hybrid and her relationship with the world of women, something that can be seen through the cosmetic or decorative aspects.

Many of the sphinxes in the Neapolitan Ovid manuscript wear headdresses (Figure 7), which, as we have seen, has been linked to their Islamic models. In addition, it may perhaps marks the fact of the extreme and perhaps dangerous femininity of the hybrids, as might be the case in the images of Circe mounting a goat (fol. 176r) and the female centaur turning her head to the sun that appear in the manuscript (fol. 80v). Both figures wear similar headdresses, establishing a formal relationship with the "canonical" sphinxes as female figures. Furthermore, the sphinxes in the manuscript that do not wear headdresses (Figure 8) have short and carefully styled hair, which poses a problem for the identification of the human figures in the Neapolitan Ovid as it seems to be common to both male and female figures.[12] However, this hairstyle gives the sphinxes a suggestive femininity, which is reinforced by the feeling that their features are marked, as if they were all wearing makeup, although it seems clear that this is merely a stylistic feature shared by all the figures in the manuscript.

The griffins, for their part, have markings on their heads as well. The most obvious are the ears, which have been part of this hybrid from the first creation of the figure in Antiquity. The Oriental version of this particular hybrid, which arrived in Greece in the 7th century B.C., added a kind of crest or horn in the form of a knob to the eagle's head, which is characterized by a strong curved beak always opened to reveal an almost serpentine tongue. The crest is a rather strange feature (Figure 10a) which disappeared in the Roman and Medieval versions (Leventopoulou 1997). The griffins of the "Napoletano" always have clearly pointed ears, along with curved beaks that are sometimes shown clearly opened (Figure 9).

However, despite the freedom of artistic composition that makes the griffins and sphinxes of the "Napoletano" a variety of customized figures, their presence and features seem to be related to the contemporary decorative motifs from Antiquity appearing on artistic objects such as textiles, ceramics, and ivory caskets as well as on architectural elements that reveal the influence of such models. On the one hand, regarding the widespread presence of the griffin in the Byzantine repertoire,[13] Andre Grabar stated that it was "un monstre acclimate de longue date dans l'ambiance gréco-latine" (Grabar 1963, p. 93), reproduced as a highly stylized motif on reliefs on church architraves, closure slabs, and decorative panels (Grabar 1963, pp. 90–99). Nevertheless, the motif was disseminated through luxury objects such as embroidered garments and textiles as well as other movable artifacts (Kalavrezou 1997). In her study of the griffin motif on Byzantine textiles, Anne McClanan

(2019) explored "the place of griffins within elite cultural *milieux*" (McClanan 2019, p. 133), the "Islamicizing" visual language used to represent these figures (Figure 11b), and more insightfully, discussed the meaning of the creature on luxury objects circulating around the Mediterranean at the time and the message of prestige they conveyed through their association with the Byzantine imperial culture (McClanan 2019).

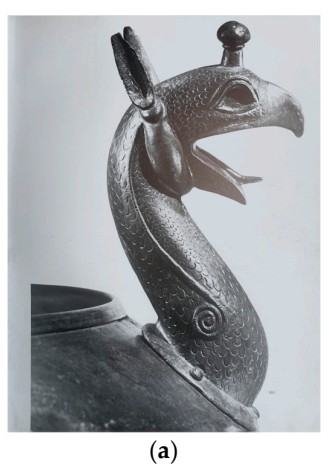
(**a**)

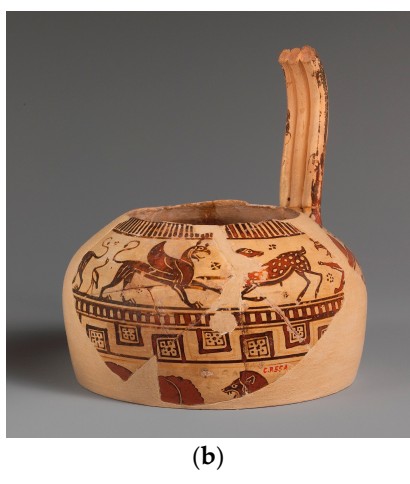
(**b**)

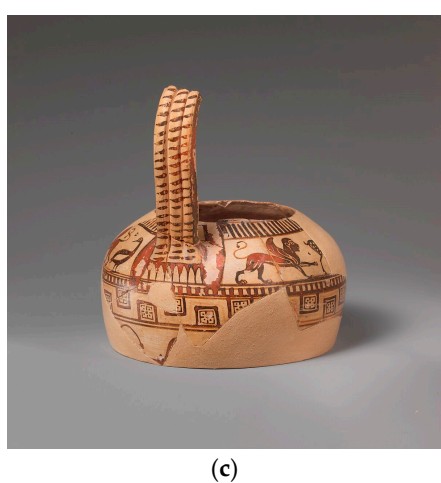
(**c**)

**Figure 10.** Ancient Greek griffins: (**a**) Griffin protome. Bronze. 7th century BC. Samos, Museum. 1239. Photo: (Hampe and Simon 1981); (**b**) Terracotta East Greek oinochoe. Attributed to the Altenburg Painter. 6th. Century BC. New York, Metropolitan Museum 74.51.365a (**c**) *Idem*, side b. New York, Metropolitan Museum 74.51.365b. Photo: Museum (Public Domain).

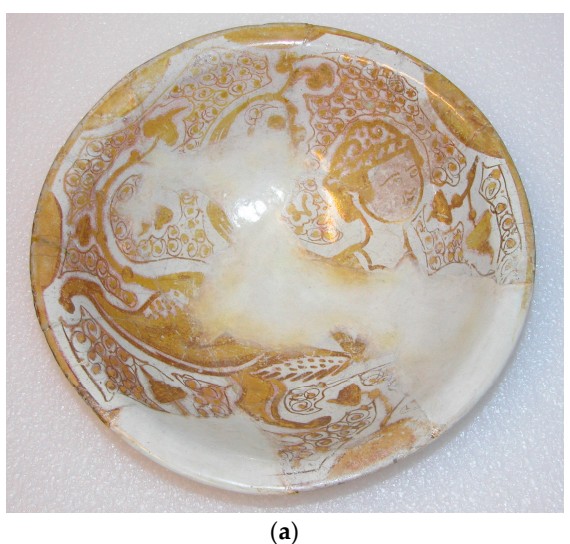
(**a**)

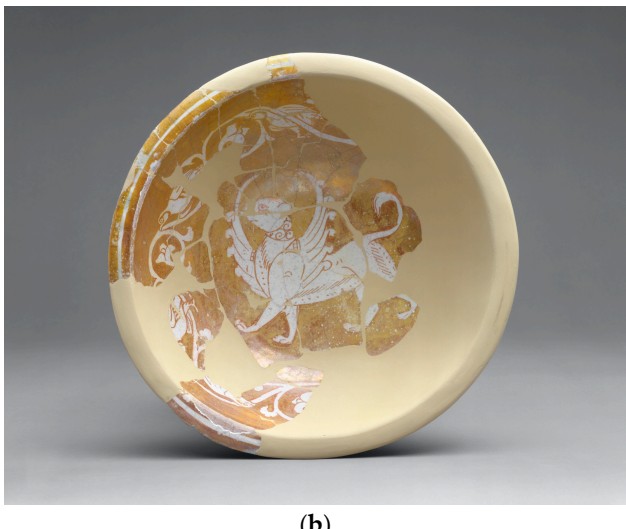
(**b**)

**Figure 11.** Islamic sphinxes and griffins: (**a**) Islamic ceramic bowl with a sphinx (second half 10th century). From Egypt. Metropolitan Museum of Art, 52.122. Photo: Museum (Public Domain); (**b**) Islamic ceramic bowl with a griffin (11th century). From Egypt. Metropolitan Museum of Art, 1970.23 Photo: Museum (Public Domain).

The figurative status of the hybrid within the Byzantine and Islamic visual cultures is extremely relevant for understanding the presence of such "canonical" and at the same time stylized griffin figures in the Barese manuscript, as they reflect how deeply rooted the decorative motif was in the Byzantine repertoire of the 11th century. On the other hand, from a formal point of view, the agile and linear griffins in the Neapolitan Ovid are close to the stylized figures embroidered on Islamic and Byzantine textiles, which nourish the griffin figures on Byzantine sculptures and mosaics. One example is the extraordinary one

from the crypt of the Cathedral of Bitonto (Figure 12), which exhibits the same characteristics: pointed ears, the entire feline body, and strong decorative vegetalization of parts of the creature, such as the wings and tail (Figure 13).

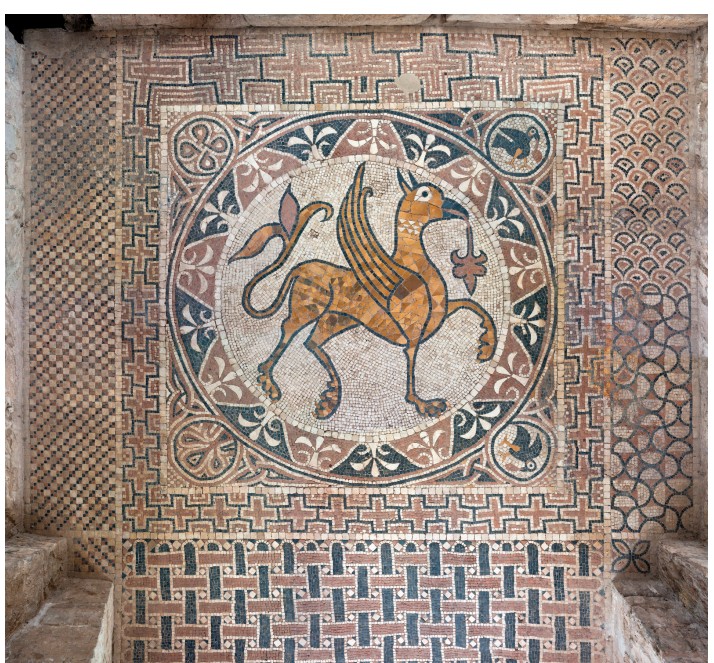

**Figure 12.** *Opus sectile* mosaic with griffin. 10–11th century. Bitonto, Cathedral. Photo: The Gourmet Gazette: https://thegourmetgazette.com/2022/08/26/the-magic-and-mystery-of-apulia/, accessed on 5 March 2023.

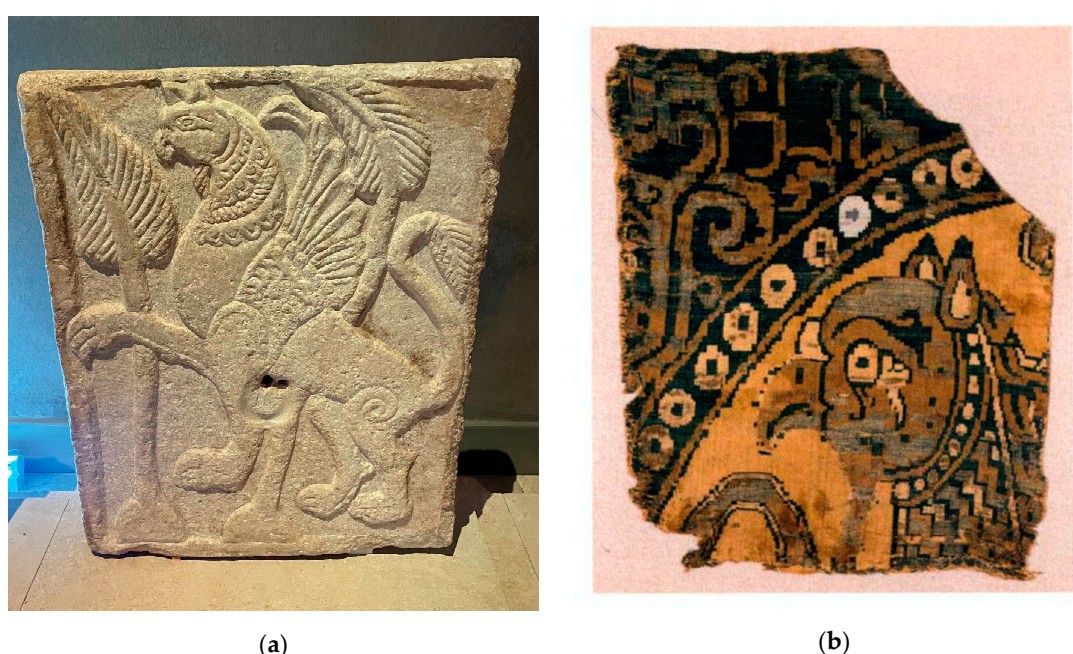

(**a**)                                                                    (**b**)

**Figure 13.** Byzantine griffins: (**a**) Church slab from Thessalonike. 10th–11th century. Thessaloniki, Museum of Byzantine Art; (**b**) Textile with griffin head. Woven silk. Byzantine workshop, 11th century. London, Victoria and Albert Museum 764–1893. (McClanan 2019, p. 138).

On the other hand, in the case of the sphinxes, the influence of the images of sphinxes from the Islamic tradition, especially that of the Fatimids, is quite conspicuous, as pointed

out by Orofino (1993, p. 14). This ancestry manifests itself in three specific features identified by Eva Baer (1965, pp. 2–3): the female heads, the short hair with a kind of pointed cap, and the leaf-shaped tail finial (Figure 11a).

As in the examples of Bitonto and Thessaloniki (Figures 12 and 13), the role of the vegetal element that characterizes the high stylization of Byzantine and Islamic decorative creations can be seen in the griffins and sphinxes of the manuscript (Figures 7–9), is installed in the forms of these hybrids and other animals such as the lions, and later becomes a constant feature in the images of the great majority of Romanesque hybrids that develop in sculptural decoration. The vegetal elements that belong to the figures and that unite the griffins and the sphinxes are a formal feature that, as already mentioned, is reminiscent of a stylistic affiliation. However, it can be intuitively assumed that they were used by the artist who decorated the manuscript for the sake of meaning and contextualization of the figures. Several of the sphinxes in the manuscript appear next to vegetal elements with leaves and flowers, which recall stylized vegetal forms (fol. 29r, 81v, and 159r; Figures 7c and 8a,b) and serve to evoke the permanent relationship of these hybrids to the vegetal element of antiquity, namely, the Tree of Life, which is sometimes reduced to a vegetal element or a simple palmette flanked by two sphinxes (Petit 2011, 2013). For example, it appears as such on the throne of Bishop Ursone in Canosa's cathedral (Figure 14), an Apulian product of Byzantine inspiration (Belli d'Elia 1975, pp. 86–91).

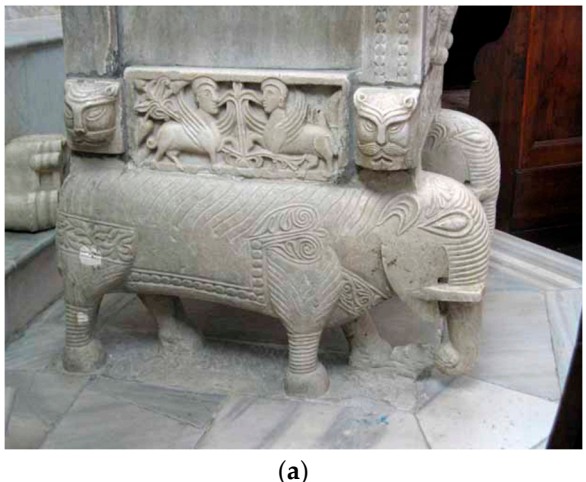 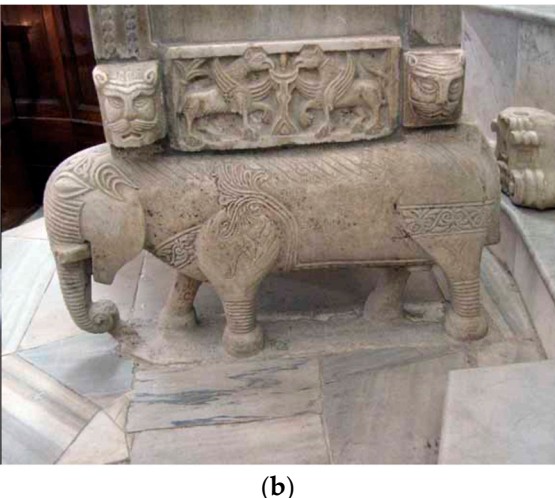

(**a**)　　　　　　　　　　　　　　　　　　　　　　　　　　　　　　(**b**)

**Figure 14.** Throne of Bishop Ursone. 11th century. Canosa di Puglia, Cathedral of San Sabino: (**a**) Right side, sphinxes; (**b**) Left side, griffins. http://www.medioevo.org/artemedievale/Pages/Puglia/Canosa.html, accessed on 5 March 2023.

In my opinion, the vegetalization of the body forms of the hybrids and other figures in the "Napoletano", which can be seen in the way that the tails or wings are represented, allows them to be objectified and perceived even more as fabulous animals without any possible existence, as the patriarch Nikephoros recalls in a text from the 9th century (Maguire 1999, p. 190). The process reveals an artistic composition of mixed beings that, in a kind of second-degree hybridization, explores the limits of transformation and fusion of natures as an encounter between the animal and vegetable kingdoms, which in the case of the sphinxes has the additional opportunity to merge with the human element.

### 3.1. Placed by the Text: The Function of Griffins and Sphinxes in the Manuscript

Most of all, it is the position and postures of the hybrid figures in the margins of the manuscript that are the most striking feature of their presence in the series of images designed to visually interpret the poem. Griffins and sphinxes are shown in action, in heraldic positions, running into the page, coming from the edge, heading toward the text or running out from it, and leaving it behind while turning their heads to look back (Figures 7–9).

Moreover, their position on the page seems to be of some importance in explaining their recurrent presence in the margins of this manuscript. On the one hand, the figures of the griffins seem to be introductory figures placed as signs next to the text. Half of the figures appear on the verso of the folios, often as the only image on the page, marking the beginning of the books (Book I, fol. 3v: Figure 15; and Book II, 16v) or the beginning of the various *fabulae* (fol. 36v, 157v, 169v, 176v), which in the manuscript are marked on the page by the presence of the Lactantian summary placed at the side and in the text by an enlarged colored initial (Figure 16).

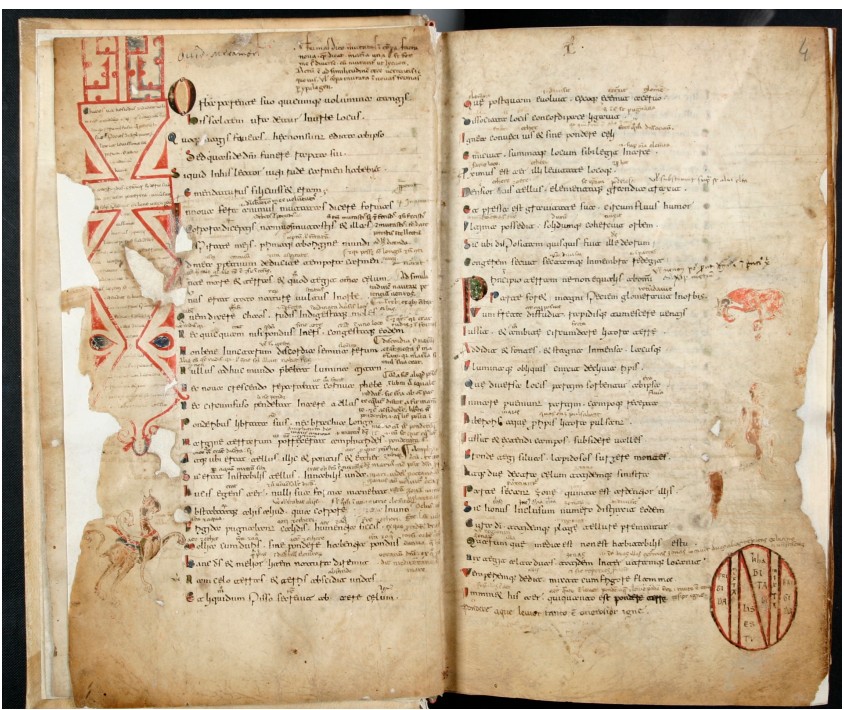

**Figure 15.** Neapolitan Ovid (BNN ms. IV F 3), fols. 3v–4r. (Public Domain).

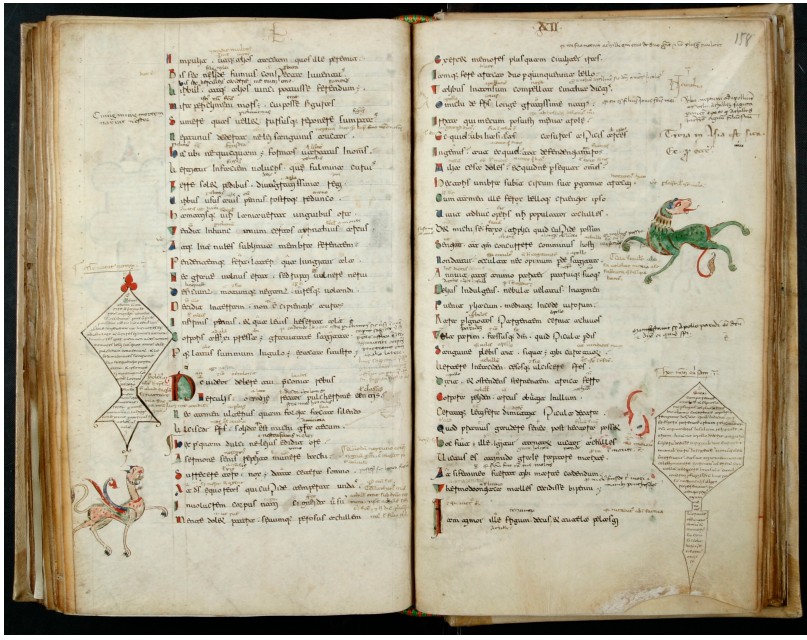

**Figure 16.** Neapolitan Ovid (BNN ms. IV F 3), fols. 157v–158r. (Public Domain).

When they are on the recto they usually look to the right, assuming a heraldic position of protection (fol. 29r, 67r: Figure 17), or seem to jump from the text to the edge of the page, as if they were leaving while looking back at the text (fol. 76r, 116r: Figure 18). Through their gestures and attitudes they seem to establish a meaningful relationship with the text, functioning as a kind of "figurative *maniculae*" that point to the important parts of the text.

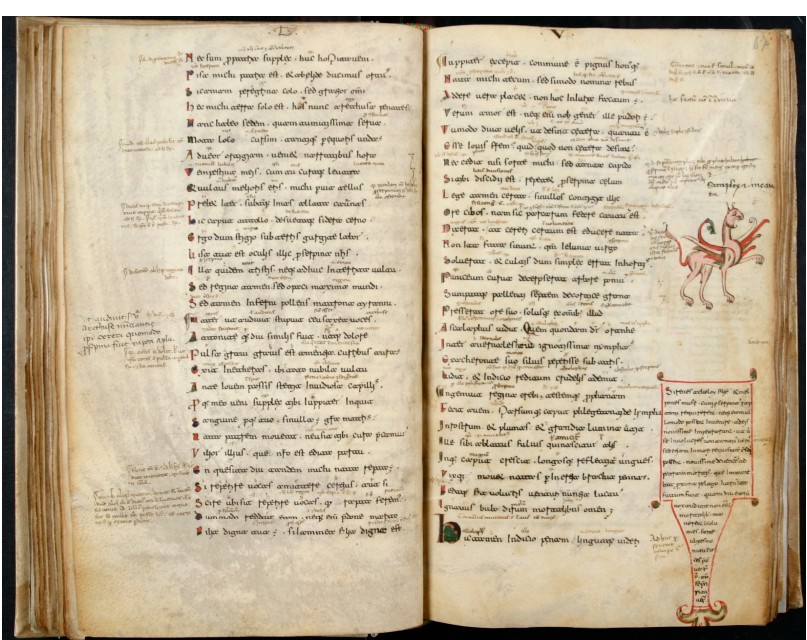

**Figure 17.** Neapolitan Ovid (BNN ms. IV F 3), fols. 66v–67r. (Public Domain).

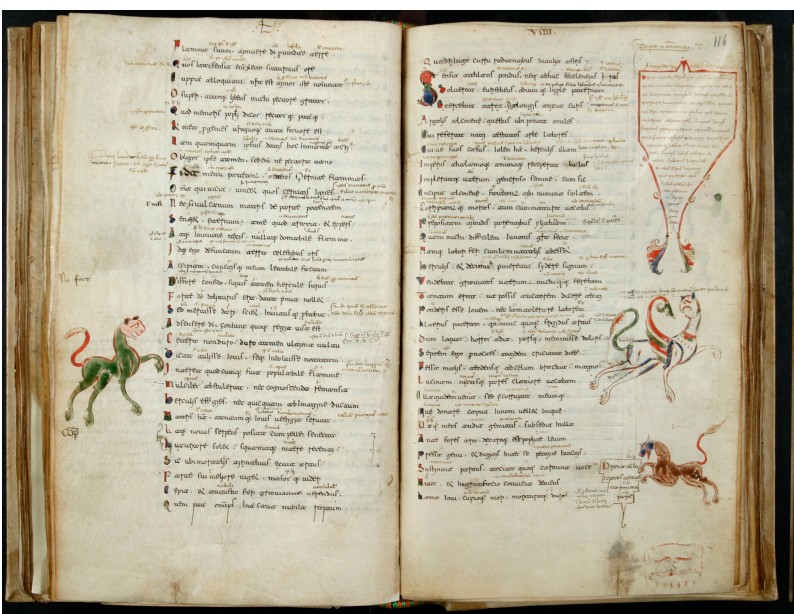

**Figure 18.** Neapolitan Ovid (BNN ms. IV F 3), fols. 115v–116. (Public Domain).

As for the sphinxes, they usually appear on the recto, either looking at the text or jumping toward it. Half of them are placed in the upper right corner, the privileged position in the process or reading (fol. 11r, 37r, 102r, 109r and 159r), and are often associated with the summaries (Figure 19). As with the griffins, they seem to function as signals in relation to the text, marking the beginning of the fabulae, giving the impression of being lively and colorful "figurative incipits". In the same way as the griffins, through their pos-

tures and gestures they assume heraldic positions or turn their heads back, and seem to mark the end of the fabulae, becoming "figurative explicits" (fols. 81v, 180r: Figure 20).

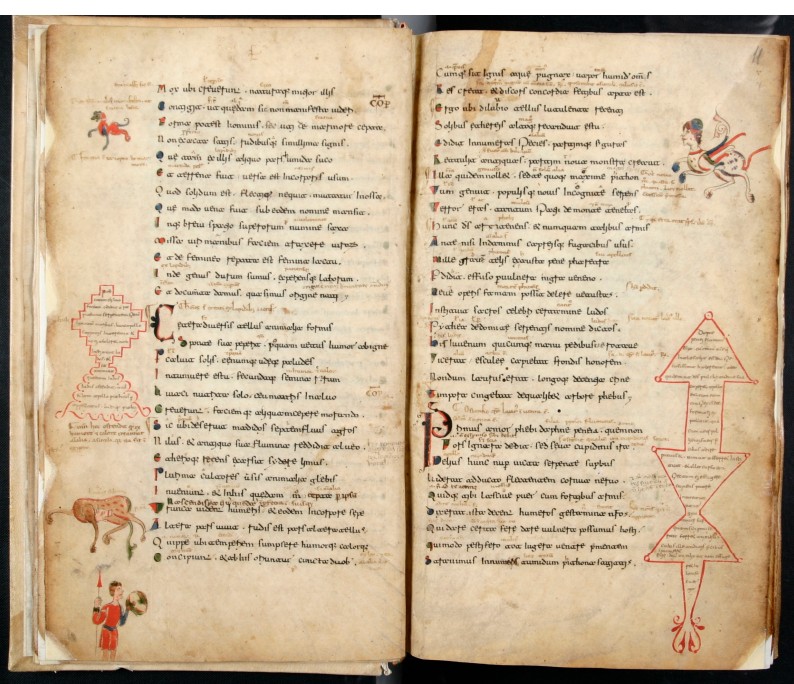

**Figure 19.** Neapolitan Ovid (BNN ms. IV F 3), fols. 10v–11r. (Public Domain).

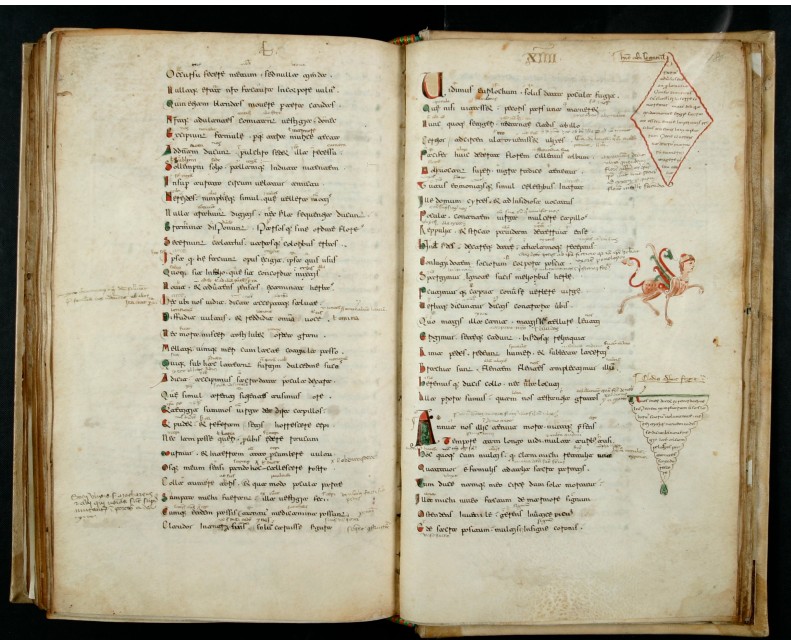

**Figure 20.** Neapolitan Ovid (BNN ms. IV F 3), fols. 179v–180r. (Public Domain).

### 3.2. Meaningful Interactions: Griffins, Sphinxes, and Other Animals

In addition to establishing a significant relationship with the material text, the griffins and sphinxes interact with other elements on the page. First, as mentioned above, the framed summaries are transformed into images which, as with many textual images, function as iconographic devices (Linardou 2017) that convey information about the content of the text as well as its structure (Vandi 2019, p. 311). Second, both hybrids interact, at times reacting to each other's presence, as can be seen on selected pages (fol. 102r: Figure 21).

Third, they share attitudes and positions in the manuscript with the various figures of felines, lions and panthers, which seem to belong to the same level of the decorative system as the hybrids and operate at a similar level of meaning. It is obvious that griffins and sphinxes share a lion's body as the basic part of their mixed nature, a feature that is obviously represented in the depiction of both hybrids. Moreover, it is the feline element—the lion's body—that stylistically unites them with the big cats depicted in the manuscript. In fact, the illuminator of the "Napoletano" used the feline body as the basis for the images of the more than twenty dogs that swarm around the margins of the manuscript, and which enter the decorative system as well.[14]

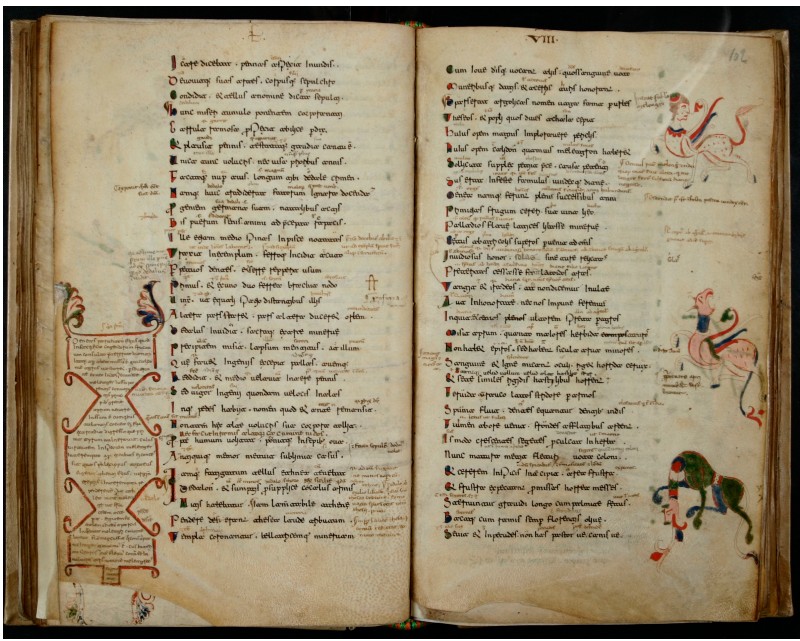

**Figure 21.** Neapolitan Ovid (BNN ms. IV F 3), fols. 101v–102r. (Public Domain).

Although the presence of lions in the manuscript is rare, it is significant for the relationship that they establish, above all, with the griffins. Of the five figures, four adopt different attitudes that correspond to the movements, gestures, and actions of the griffins and sphinxes already examined. They are shown in heraldic positions, with one paw raised, moving towards the text or away from the text and looking back; all four share both their formal features and most of their gestures and postures with the hybrids (Figure 22).

In two instances (fol. 76r and 158r), the special relationship established between the griffins and lions in the manuscript could be defined as "systematic", which is the case for two reasons. On the one hand, as has already been pointed out, both figures seem to belong to the collection of figures that refer to the text itself rather than to its content. Despite the story told in the verses written beside them, the griffin and the lion on fol. 157v–158r (Figure 16), by drawing attention to the text and to each other, both create a kind of figurative chiasm that plays with the framed summaries that introduce the two fabulae narrated in this text from Book XII. The griffin is on the verso, at the bottom of the page below the summary, raising one paw in an attitude of worship or protection towards the text, acting as a kind of "figurative incipit" that reinforces the textual rhythm established by the marginal Lactantian summary and the initial ornament (the large green N). At the top of the recto, a green lion moves toward the text, also raising its paw but looking outward from the page. The figure is placed along the last verses of the *fabula* beginning on fol. 157v and above the framed summary that marks the beginning of the next *fabula*. The raised paws of both figures obviously refer to the text as an object of worship or protection, and the direction of the lion's head, looking outwards from the page, again seems to suggest something in relation to the text, this time acting as a kind of "figurative explicit".

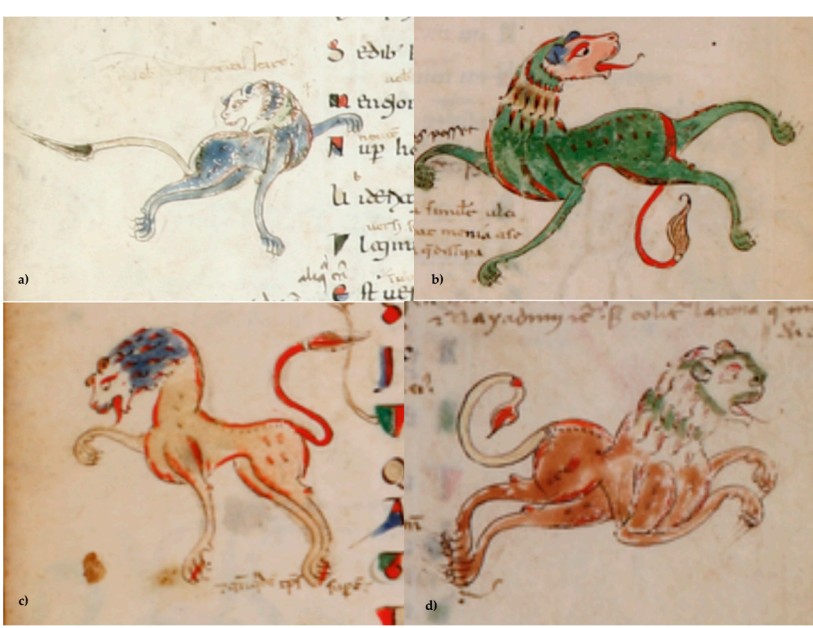

**Figure 22.** Neapolitan Ovid. Lions: (**a**) fol. 25v; (**b**) fol. 158r; (**c**) fol. 177v; (**d**) fol. 76r.

On the other hand, this formal relationship between griffins and lions takes on a different status on another page of the manuscript, where the two figures, here located in the margin of the recto (fol. 76r), leap towards the edge of the page, as if they were advancing on the text in a posture of defense or attack (Figure 23). Their posture urges the viewer to turn the page in order to find the target of both animals, thereby re-establishing a relationship with the text, the page, the *mise-en-page*, and the manuscript as a whole, in the sense pointed out by Vandi (2019, p. 314). Nevertheless, apart from the matching attitudes, I would like to emphasize the formal association established between the two figures, griffin and lion, which in my opinion effectively evokes the frequent representations of both animals together on objects and spaces in Byzantine art, which will be addressed below.

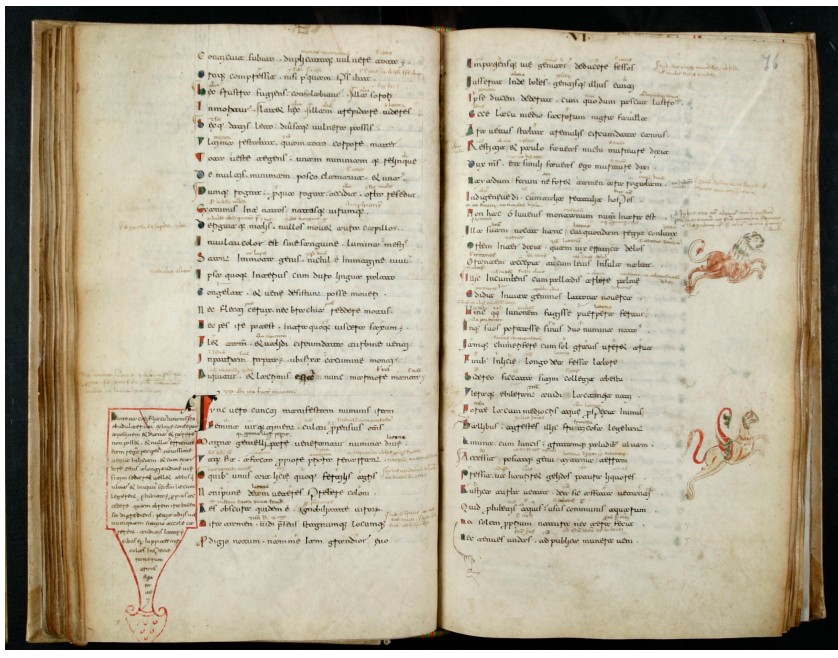

**Figure 23.** Neapolitan Ovid (BNN ms. IV F 3), fols. 75v–76r. (Public Domain).

## 4. An Inherited Repertoire: Images for Significant Decoration

The presence of griffins and sphinxes in the Bari manuscript has been interpreted as part of the influence of the cultural *milieu* of Byzantine and Islamic influences in the period during which the codex was copied and illustrated (Bertelli 1975; Orofino 1993, 1995, 1998, 2020; Vandi 2019). As is the case for Byzantine and Islamic griffins and sphinxes, the hybrid figures in the manuscript maintain the formal characteristics and meaning of the mixed beings of Antiquity. On the one hand, on the basis of the same composition, those few formal changes are revealed in the transformation of animal limbs such as wings or tail into vegetal elements and in the decorations that embellish the head, neck, and body. On the other hand, the griffins and sphinxes retain their attitudes and postures, several of which could be assimilated to heraldic positions of protection or reverence. Particularly noteworthy is the posture with one front paw raised; this is a characteristic of these figures which, like the markings on the heads, has been handed down from antiquity and was already clearly encoded in the depictions of griffins and sphinxes in the Archaic Greek world (Winkler-Horaček 2011c; Böhm 2020). This gesture has an obvious meaning related to protection or reverence, which endows the hybrid with values that are partly apotropaic and partly related to the power of defense and the guardianship of sacred or sacralized elements or spaces. In the Byzantine artistic repertoire, the meaningful function of reverence and protection from Antiquity is revealed through this conspicuous posture of the two hybrids, represented in architectural reliefs in symmetrical pairs flanking a vegetal element, a palmette, or a bush or tree, as in the throne of Bishop Ursone in the Cathedral of Canosa (Figure 14). Thus, the presence of griffins and sphinxes on funerary monuments, church iconostases, and various objects is intended as a protective or worshipful device, as conceived in the Oriental world and developed in the Greek world (Papalexandrou 2016).

Most striking and revealing of the value of the presence of griffins and sphinxes in the "Napoletano" is the reunion of griffins, sphinxes, lions, and panthers, which seem to have the same functional status within the decorative system, defined here as a group of figures that develop their relationship with the text and the page rather than with the thematic content of the poem. All four types of hybrids and animals have a long history in the visual arts that can be traced back to the 7th century BC, the so-called Orientalizing period (Childs 2003), when images of griffins, sphinxes, and sirens arrived in Greece along with lions and panthers as part of a repertoire of motifs of Oriental origin (Burkert 1992, p. 19). Keeping their original meaning as representatives of the liminal character of the dangerous world of wild nature, mixed beings and roaring wild animals began to share space as a stable figurative group on the decorated bands of Orientalizing Corinthian pottery, developing the status of decorative devices filling the marginal spaces of various objects and supports (Winkler-Horaček 2000, 2011a, 2016). Although they retained their oriental features and underwent only slight formal changes, these figures of enigmatic hybrids and fierce predators lost some of their terrifying character. Thus, on the surface of Greek pottery, griffins, sphinxes, sirens, lions, and panthers began to alternate in the figurative scenes, both among themselves and with other tame herbivores, composing what R. Olmos has called "una teoría de animales," a simple parade of animals in peaceful coexistence and paratactic relationship (Olmos Romera 2003, p. 108). This strange zoo made up of hybrids, wild animals, deer, bulls, and birds composes a fabulous natural world that serves to decorate the margins with a representation of nature as a meeting point between the human and the divine (Díez Platas 2023); this is a marginal world, alien to civilization, endowed with a variety of meanings and related to all kinds of transits, including death (Tsiafakis 2003).

In a suggestive article on the heritage of Antiquity in medieval Puglia, Belli d'Elia (1998) suggests the existence of a continuity of repertoires and meanings of forms and figures of Magna Graecia in the artistic manifestations of southern Italy, specifically in places like Bari, which is the context for the creation of the Neapolitan manuscript of Ovid. This is an inheritance that the author describes as "inconsapevole," unconscious, in the sense that there seems to be no conscious transmission from ancient Greece to the medieval environment of southern Italy. In Belli d'Elia's opinion, however, this relationship, which seems to

be verifiable, takes place through the mobile objects that have survived from the past, such as Orientalized Greek pottery (Figure 10b,c), which arrived in Magna Graecia from the very beginning, along with it this repertoire of figures that formed the standard decoration of numerous vessels and the local production of Apulian vases (Figure 24), which shared the same repertoire with the mainland and Oriental Greeks (Farioli Campanati 1996).

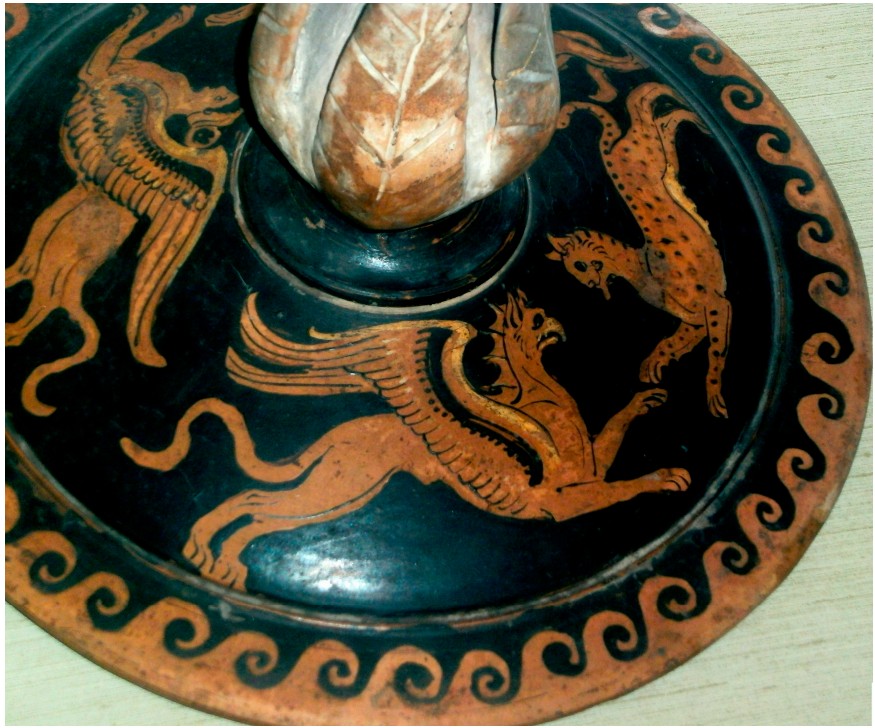

**Figure 24.** Greek pottery lid with griffin, sphinx, panther, and lion, 4th century BC. Apulia. Compiègne, Musee Antoine Vivenel. Photo: Wikimedia commons: https://commons.wikimedia.org/wiki/User:Zde/Griffin?uselang=fr#/media/File:Griffin_greek_petery.JPG, accessed on 5 March 2023.

As is the case with the Barese manuscript itself, the portable objects—pottery, textiles, ivory caskets, and oliphants—that transmitted this meaningful decoration were an exercise in luxury, and the materials and decoration show the refined taste for color and form that is expressed in an extreme way in the Byzantine world (Dauterman Maguire 1997; Cutler 1998; Maguire 2004; McClanan 2019). The rich decoration of the manuscript, including the presence of the Greco-Oriental repertoire that plays its role in the margins of the pages and the "Islamicizing" decoration, such as the pseudo-Cufic letters and the *rumi* leaves, reveals a decorative coherence similar to that of certain objects with complex ornamentation from the hybrid cultural world of Southern Italy in the 11th century: a "meaningful mingling", using the words of Alice Walker from a suggestive work on an extraordinary vase from the Treasure of San Marco (Venice) that reflects both luxury and cultural hybridity (Walker 2008).

A particularly significant case may serve to illustrate this relationship between the decorative system of the Ovidian manuscript and objects that may have served not only as models for the transmission of motifs but above all as material correlates, i.e., as luxury objects full of meaning and protected by a specific decoration: the incense burner or lamp in the shape of a domed building belonging to the Treasure of San Marco in Venice (Evans and Wixom 1997, p. 251, nº 176). This incense burner has the shape of an architectural structure with five domes, which seems to imitate the structure of the Venetian church itself (Figure 25). The domes are pierced with vegetal decorations to let in either incense smoke or light, and the walls are fitted with doors as if it were a real building. The lower part of the wall is decorated with repousse panels that surround the entire body of the incense

burner. The decorations, symmetrically arranged as a series of metopes, depict a lion and a griffin followed on either side by sirens, a centaur, and other scenes of classical flavor. The griffin and the lion, shown in the repeated pose of protection or reverence, flank the double door that opens at the front of the building. On the doors, the figures of a warrior and a woman represent the personification of two virtues, Courage and Intelligence, as indicated by the accompanying Greek inscriptions. The decorative elements of the incense burner—roundels, palmettes, scroll bands and rosettes—are completed by the human and feline faces that hang above the doors and on either side of each handle.

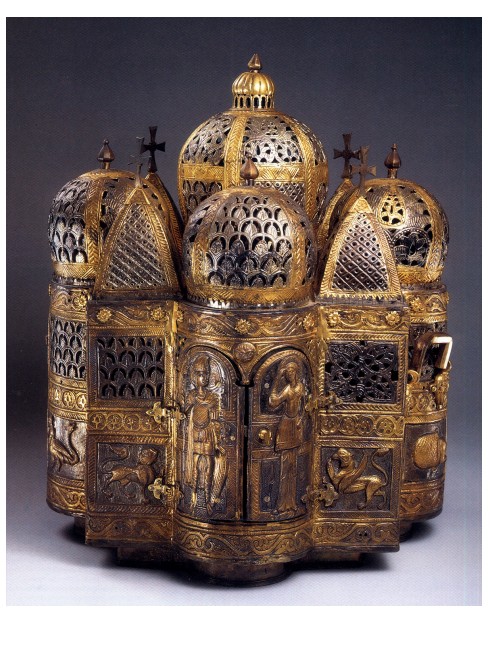 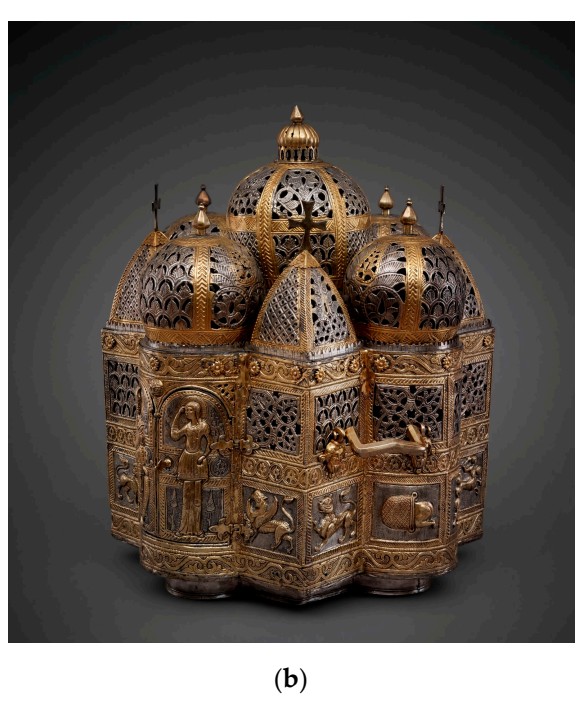

(**a**)　　　　　　　　　　　　　　　　　　　(**b**)

**Figure 25.** Incense burner in the shape of a domed building. Byzantine. 11th century. Silver, partially gilded. Venice, Procuratoria di San Marco. (**a**) Front; (**b**) Left side. (Evans and Wixom 1997, nº 176).

Together with the repertoire of ancient images, the whole decoration of this unique luxury object can be compared, in my opinion, to the composite decoration of the Neapolitan Ovid, made up of vegetal decoration, incipient mythological scenes, animals, and hybrids—centaurs, sirens, griffins, sphinxes, and lions—which flank the text, adorn it, and function at the same time as significant decorative and protective devices. As already suggested, I would argue that the Barese manuscript, like the magnificent incense burner, can be considered a luxury object that illustrates the possibilities of the image to embellish a significant space and convey specific visual and cultural messages.

## 5. Conclusions

In the Neapolitan Ovid, the "canonical" and decorative griffins and sphinxes, like the rest of the figures that crowd the margins, are infected by the transforming vitality proclaimed in the poetic content of the *Metamorphoses*. As a result, they are activated and come to life to assume a new position and a new function. The obvious hybrids become involved in the game board composed in the manuscript and in the underlying space of the decorated page, surrendering themselves to the game that unfolds through the pages inside the codex.

Ignoring the metamorphoses that take place around them in the images that evoke and provoke the content of the poem, the griffins and sphinxes interact with each other and with the other figures of their figurative realm: the lions, the hunting panthers, and the pack of dogs unleashed from within the Ovidian text. This systematic repertoire of living hybrids and felines, images of ancient Magna Graecia, performs the same function

of adornment and custody in the Apulian manuscript as that observed on precious objects and artifacts or in the decorated liminal spaces of churches.

In the manuscript space, however, the griffins and sphinxes, as solemn guardians and enigmatic protectors of different realities, exercise their function over a different element: the text of the *Metamorphoses*. The poem of changes, full of a life that overflows and manifests itself in constant transformation, is materialized in the Neapolitan Ovid in a text composed of colored letters and images that emerge from the writing itself and overflow into the decorations in the margins. The ancient hybrids, in their canonical form and only partially transformed by the stylistic phenomenon, are attracted by the text and burst into the margins of the page to mark and protect it, like an element "sacralized" by their presence and their attitude. The migration of these fabulous hybrids and the transformation of their function together suggest another step in the many metamorphoses that take place under the influence of the virtual reservoir of sounds, lights, colors, and shapes that constitute the Ovidian poem. In this extraordinary manuscript, the griffins and sphinxes become the guardians of a new space, a new "tree of life", this time a living construction of words.

**Funding:** This research received no external funding.

**Data Availability Statement:** No new data were created or analyzed in this study. Data sharing is not applicable to this article.

**Acknowledgments:** This research is partly related to the project entitled, *Manuscritos bizantinos iluminados en España: obra, contexto y materialidad-MABILUS* (MICINN-PID2020-120067GB-I00) (https://mabilus.com), in which I am participating as member of the working team.

**Conflicts of Interest:** The authors declare no conflict of interest.

## Notes

1. On the manuscript: (Cavallo et al. 1998). Online information and digitized copy: www.loc.gov/item/2021667680/, accessed on 5 March 2023. Digital file at the Biblioteca Nazionale di Napoli: http://digitale.bnnonline.it/index.php?it/149/ricerca-contenuti-digitali/show/89/, accessed on 5 March 2023. An exhaustive actualized bibliography on the codex (1990–2020) at https://bmb.unicas.it, accessed on 23 February 2023.

2. The codex is an Apulian product that once belonged to the Neapolitan convent of San Giovanni a Carbonara; it was most likely copied around 1090 at the *scriptorium* of the monastery of San Benedetto di Bari: (Orofino 1993, p. 15). The presence of the manuscript at the Montecassino Abbey is attested in the 12th century. By a note of ownership on fol. 201 we learn that during the 15th century the codex was in the hands of a Neapolitan bishop, Girolamo Seripando, who in his will bequeathed his own library to the Neapolitan monastery of San Giovanni a Carbonara. In 1800, the codex became part of the former Royal Library, now the National Library of Naples. For the history of the manuscript, codicological description, and paleographic analysis: (Magistrale 1998).

3. There are twenty-five illuminated manuscripts of the *Metamorphoses*, dating from the eleventh to the fifteenth century. On the illustration of the *Metamorphoses* in the Middle Ages: (Orofino 1995; Buonocore 1996; Lord 2011; Toniolo 2018).

4. The first book has no capital initial (fol. 3v), and the XVth book is a later addition (fols. 189–201).

5. In her survey of the medieval image of the *Metamorphoses*, Carla Lord devotes a brief review to the Neapolitan manuscript (Lord 2011, pp. 257–59). Recently, Loretta Vandi revisited the manuscript in a study on illuminated manuscripts in the Xth and XIth centuries (Vandi 2019). On some aspects of the mythological figuration, see also: (Pesavento 2014; Venturini 2014, 2018).

6. "( . . . ) in the margin of the Ovidian [text] the medieval novel of mythology".

7. "These ambiguities, redundancies and deficiencies remind us of those which doctor Franz Kuhn attributes to a certain Chinese encyclopedia entitled The Celestial Emporium of Benevolent Knowledge. In its remote pages it is written that the animals are divided into (a) belonging to the Emperor, (b) embalmed, (c) trained, (d) piglets, (e) sirens, (f) fabulous, (g) unleashed dogs, (h) included in this classification, (i) trembling like crazy, (j) innumerables, (k) drawn with a very fine camelhair brush, (l) et cetera, (m) those which just broke the vase, (n) those which from a distance look like flies."

8. In addition to the classical centaur representing the zodiac sign of Sagittarius in the vision of Phaeton (fol. 20) and the three types of sphinxes listed below (see note 9), there are ten other hybrid figures in the manuscript: (1) fol. 54r: a medieval sea-dragon reminiscent of the Greek *Ketos* (*Cetus*) used to represent the metamorphosis of Cadmus into a snake (Orofino 1993, p. 8); (2) fol. 80v: a bovine centaur with a mottled body and female torso wearing a headdress that turns her head toward the sun; (3) fol. 101r: a kind of centaur with a (female) human head embedded in a mottled bovine body that is one of the medieval raiments for the Minotaur; (4) fol. 110r: a mixed being with a human (female?) upper body embedded in a bovine forepart, from which a large

and multicolored serpentine tail grows, which also wears a headdress; (5) and (6) fol. 158v: a very similar figure, reminiscent of the Minotaur on fol. 101r, with a human head embedded in a bovine body with mottled skin, is accompanied by another kind of hybrid creature with a human body and a dog's head, which can be identified as a cynocephalus, armed with a shield; (7) and (8) fols. 159r and 159v: two different sirens or harpies, characterized by large claws, composed of a female head and a bird's body (Leclercq-Marx 1997). The first, colored, is an original miniature, while the second, simply an ink drawing, seems to be a copy of the former made by the author of the notes in this margin of the manuscript; (9) fol. 166r: a male hybrid, half-man with beard and moustache and half bull with mottled skin. The torso of the figure is bent towards the earth, giving the impression that the figure is a rare form of quadruped reminiscent of the *hippopodes*, similar to a satyr or the image of the god Pan. (Leclercq-Marx 2021, pp. 213–14), and (10) fol. 179: a mixed being, very similar to the one in fol. 110r, although it lacks the headdress and in addition has a wing very similar in shape and color to the decorative wings of the "canonical" griffins and sphinxes.

9    This is especially true for the female figures that combine human and feline or quadrupedal features, the three hybrid figures that could be considered sphinxes in composition but stylistically differ from the uniform figures of the ten sphinxes listed above: the first is the small figure of a red lion with an apparently female human head and a vegetal tail that appears on fol. 5v; the second is the clearly male bearded hybrid of fol. 104r, with a mottled feline body and a leonine tail; and the third is the small hybrid figure with an apparently female head and a pointed cap that hangs in the air at the margin of fol. 159r. Its red body is partly feline and partly bovine, according to the different shapes of its front hooves and hind paws; it also has a short tail, which is not leonine, and no wings.

10   On the Sphinx in Antiquity (Egypt, Near East and Greece): (Dessenne 1957; Demish 1977; Kourou 1997; Winkler-Horaček 2011b).

11   For a general overview on the griffin: (Brandenburg 1983) (Antiquity and Middle Ages) and (Leventopoulou 1997) (Classical Antiquity).

12   See for example the hair style of the figure identified as Jason (fol. 82r). Orofino identifies as male figures the small human heads that grow from the vegetal ribbons of some capitals (Orofino 1993, p. 5).

13   On griffins in Byzantine art: (Ćurčić 1995; McClanan 2019). On griffins in Byzantine sculpture: (Grabar 1963, pp. 90–99; Grabar 1976; Vanderheyde 2007, 2020; Skoblar 2020).

14   The same can be said of the numerous unleashed dogs that swarm across the manuscript, though we do not consider them here.

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
