# Peer review of "Guardians of the Text: Griffins and Sphinxes in the Neapolitan Ovid (BNN ms. IV F 3)"

_arts_

Round 1
Reviewer 1 Report
The article proposes an original reading of two elements of the rich illustration of the Neapolitan Ovid (Bari, Apulia, late 11th century, early 12th century), a manuscript of great importance because it is the oldest witness to the iconography of Metamorphosis in the Middle Ages. In fact, the presence of sphinxes and griffins on the margins of the pages, which until now had not been explained, is convincingly interpreted in relation to the text and the mise en page rather than to the thematic content of the text, as "figurative maniculae" signaling important passages (it would be interesting to verify which ones), "figurative incipit" and "figurative explicit," that is, as 'mnemonic devices' .
Also significant is the relationship to the figurative tradition (and role) of sphinxes and griffins from the ancient world to the Byzantine, Islamic, and Apulian Romanesque art.
The article is very clearly written, is methodologically correct, is presented in a well-structured manner, and reveals a thorough knowledge of the bibliography related to the manuscript and the iconography of the subjects covered. The set of images is appropriate.
ll. 75, 751 and note 5: Loretta, not Loreta
Author Response
Thank you very much for the review and the positive comments. I have already made the changes that you suggested.
Reviewer 2 Report
The article is straightforward and focused on a clear question, developing its arguments in a compelling way. Not being a specialist in manuscripts specifically, I appreciated the way the author engages the discussion of how these images were transmitted: could the illuminators have seen Roman/Greek objects in the vicinity? Did the "marginal" place of griffin images in Greco-Roman visual culture (often placed on the sides of sarcophagi) further fuel the marginal place of these figures in the manuscript? These questions reminded me of the debates over the origins of certain motifs of ancient origin in Romanesque churches in sites where the Roman presence was abundant (Arles and Saint-Gilles du Gard being the most emblematic sites). The question of what could really be seen and of "antiquarian" culture in the 11th is obviously highly complex, but could be developed slightly more in the text. The issue of the more immediate transmission of these motifs from "Byzantine" and Islamic contexts is convincingly resolved.
Considerations of the actual cultural hybridity of these images, which are after all intended for a reduced audience, could also be developed further, as could the interaction between these images and the pseudo-scriptures (and their meaning). But for this last point, it is perhaps an element that would distract from the main argument of this article, which deserves to be published.
Author Response
Thank you for your positive comments and suggestions. I think it is reasonable to assume that ancient Greek objects, such as decorated pottery, were available and that certainly contemporary objects transmitted the motifs, as Belli d’Elia (1998) suggests, showing the existence of a continuity of repertoires and meanings of forms and figures of Magna Graecia in the artistic manifestations of southern Italy.
Regarding the marginal position of griffins and sphinxes in the manuscript, of course it can be seen as a continuation of the position they had in Greek and Roman visual culture, because the position of these hybrid creatures was essential to show their function and meaning.
I think I tried to show this continuity in my article, as well as the concrete question of the possibility of the 'antiquarian' view, but the main topic of the article was to explain the meaning and function of the Graeco-Roman hybrids in the manuscript within the lavish and complex decoration developed to accompany the text of the Metamorphoses.